# Optimal Scheduling Method of Controllable Loads in Smart Home Considering Re-Forecast and Re-Plan for Uncertainties

**Akihiro Yoza [1],\*** , **Kosuke Uchida [2]**, **Shantanu Chakraborty [3]** , **Narayanan Krishna [4]** , **Mitsunaga Kinjo [1]**, **Tomonobu Senjyu [1]** and **Zengfeng Yan [5]**

[1] Department of Electrical and Electronics Engineering, University of the Ryukyus, 1 Senbaru, Nishihara-cho, Nakagami, Okinawa 903-0213, Japan; mitsu18@tec.u-ryukyu.ac.jp (M.K.); b985542@tec.u-ryukyu.ac.jp (T.S.)

[2] Department of Electrical and Electronic Systems Engineering, Ibaraki University, 4-12-1 Nakanarusawacho, Hitachi, Ibaraki 316-8511, Japan; kosuke.uchida.ee@vc.ibaraki.ac.jp

[3] Department of Earth Science Faculty of Science, The University of Melbourne, 187 Grattan Street, Carlton, VIC 3053, Australia; shantanu.chakraborty@unimelb.edu.au

[4] Department of Electrical and Electronics Engineering, SASTRA Deemed University, Thanjavur 613401, India; narayanan@eee.sastra.edu

[5] School of Architecture, Xi'an University of Architecture and Technology, Xi'an 710055, Shaanxi, China; xazfyan@126.com

\* Correspondence: orihika085584h@gmail.com

**Abstract:** Renewable energies (REs) such as photovoltaic generation (PV) have been gaining attention in distribution systems. Recently, houses with PV and battery systems, as well as electric vehicles (EV) are expected to contribute to not only the suppression of global warming but also reducing electricity bill on the consumer side. However, there are numerous challenges with the introduction of REs at the demand side such as the actual output of REs often deviating from the forecasted output, which causes fluctuation of the power flow and this is challenging for the distribution or transmission system operator. For this challenge, it is expected that smart grid technology using controllable loads such as a fixed battery or EV battery, can suppress fluctuation of power flow. This paper presents a decision method of optimal scheduling of controllable loads to suppress the fluctuation of power flow by PV output in the smart home. An optimization method to cope with uncertainties such as variability of PV power and effective forecasting methods are considered in the proposed scheme. In order to decrease the expected operational cost and to validate the robustness for the uncertainty's optimization approach, statistical analysis is executed for the optimal scheduling scheme. From the optimization results, the proposed methodology suppressed the fluctuation of power flow in the smart home and also minimized the consumer operational cost.

**Keywords:** smart home; optimal scheduling; photovoltaic generation; forecast; uncertainties

## 1. Introduction

To prevent global warming and the exhaustion of fossil fuels, the introduction of renewable energies such as the photovoltaic generation (PV) and wind energy generation (WG), has been gaining attention in power systems. Moreover, an energy storage system (ESS) such as a battery is expected to play a significant role in stabilization of power flow fluctuation caused by the varying power output of the renewable generators due to unstable weather conditions [1,2]. Recently, in the residential side, the introduction of PV and battery (either electric vehicle or small fixed type) system is gaining attention, since the PV and batteries can contribute to reduction of peak load. In order to reduce peak load and

electricity cost in demand side, an optimal scheduling of household appliances is required and have been investigated [3–6]. Where, it is very important to consider uncertainties of PV output in the day-ahead scheduling, because the actual PV output often deviates from forecasted value.

In order to cope with the uncertainties imposed by the forecasting error of PV output, a stochastic optimization programming such as a scenario-based approach has been studied and used to solve the scheduling optimization. The scenario-based approach can deal with the forecast error deriving various scenarios based on a certain probability density function and can minimize the expected cost under certain conditions [7–10]. The literature [9] provides a stochastic model for an optimal scheduling of REs and thermal units in micro grids to maximize the expected value of profit in electrical market. In the paper, the uncertainties of PV output are considered using scenario-based approach in which the scenarios of PV is generated on the basis of the probability distribution function of forecasted error. In Reference [10], a stochastic programming model for an optimal scheduling of distributed energy resources such as ESS with PV system is presented. The paper is taking into account the uncertainties for PV output and presents the setting of the probability of solar irradiance in detail. The literature [7–9] addresses the scenario-based method's problem which the calculation time increases as the number of scenarios increase. This problem has been improved by a countermeasure such as reduction of scenario [7–9].

Another approach that focuses on a forecasting method for uncertain REs output, is proposed in References [11–13]. Reference [12] describes the significance of the accuracy of the forecasted value using the re-forecasting model in order to maximize the benefit in electricity market. Reference [13] presents dispatch scheduling of generators and ESS where the probabilistic forecast method is used in order to deal with the uncertain REs output. Additionally, the data used to forecast is updated every hour and the forecast accuracy can be improved.

From above background, the uncertainties caused by the forecasted error of PV output should be considered in the scheduling of controllable appliances such as batteries in residential side. Moreover, the consideration of re-forecast is necessary to improve forecast accuracy as well. In our previous work [14], we proposed an optimal scheduling method by applying the scenario-based approach in a smart home, however re-forecast and re-plan has not been considered in the optimization process in the paper. It is expected that operational cost can be even reduced using the latest data which are atmosphere, humidity, and so forth, and needed to forecast PV output. These factors are generally updated every several hours [11].

The objectives of this paper are to minimize the expected operational cost for the resident using controllable loads such as batteries and electric vehicle (EV) installed in a smart home while also considering uncertain PV output. In this paper, it is assumed that the power company provides a suitable command value for power flow in the smart home. Then, the optimum schedule of controllable loads such as batteries is solved with using tabu search (TS) algorithm considering uncertainties for finding charge or discharge control of a small fixed battery and EV systems according to the condition of the PV generation power, where the PV power output is re-forecasted every several hours by the neural network (NN) [15] forecasting method. In addition, the scheduling in the optimization process is re-planned every several hours according to re-forecasted PV output. An additional purpose of this paper is to reveal how much the expected operational cost can be reduced by introducing re-forecast and re-plan in addition to consideration of the uncertainties.

This paper provides a robust optimal scheduling decision method of controllable loads which are battery, EV and heat pump (HP) in a smart home, which can address the uncertainties by applying scenario-based approach in a stochastic optimization. Additionally, re-forecast and re-plan are considered in the optimization process to further decrease the expected operational cost. In order to verify the effectiveness of the proposed method, the results obtained in four different cases are compared and discussed in the simulation results. Finally, the optimal schedules obtained in the four cases are tested for the PV output involving uncertainties in order to confirm the robustness. This is done using Monte Carlo simulation for the various scenarios. The comparison of simulation results

obtained in four cases is stated and examined. The statistical analysis mentions the significance of consideration of the uncertainties, re-forecast and re-plan in the scheduling of the controllable loads, and it implied that the capacity of batteries can be reduced by the proposed optimization method in the statistical analysis. The effectiveness of the proposed method is validated with using MATLAB®.

The rest of the paper is organized as follows. Section 2 describes the power system model involved in the smart home. Section 3 presents a formulation of the optimization problem with the TS algorithm, including the uncertainties. Section 4 discusses simulation results which are derived from the optimization method. Furthermore, the results are statistically confirmed to verify the effectiveness of the optimal solution by using Monte Carlo techniques. Section 5 presents the conclusion and future research avenue.

## 2. Power System Model

In this section, the composition of appliances installed in the smart home is explained. The smart home system is expressed in Section 2.1 and the PV system and the solar collector (SC) system are expressed in Sections 2.2 and 2.3, respectively.

### 2.1. Smart Home System

The smart home system model is depicted in Figure 1, which was proposed by our previous work [1,3]. The smart home is assumed to be equipped with PV panels, SC, HP, residential battery and EV system. The HP, battery and EV systems are used as controllable loads in the smart home. As shown in Figure 1, $P_{I_t}$ is power flow of interconnected point in the smart home, $P_{L_t}$ is power consumption except for controllable loads, $P_{B_t}$ is charge/discharge power of the battery, $P_{EV_t}$ is charge/discharge power of EV, $P_{PV_t}$ and $P_{HP_t}$ are PV output and power consumption of HP, respectively. The power flow of interconnected point in the smart home at each time index $t$ is expressed in following equation.

$$P_{I_t} = P_{L_t} - P_{Bt} - P_{EVt} - P_{PVt} + P_{HPt} \tag{1}$$

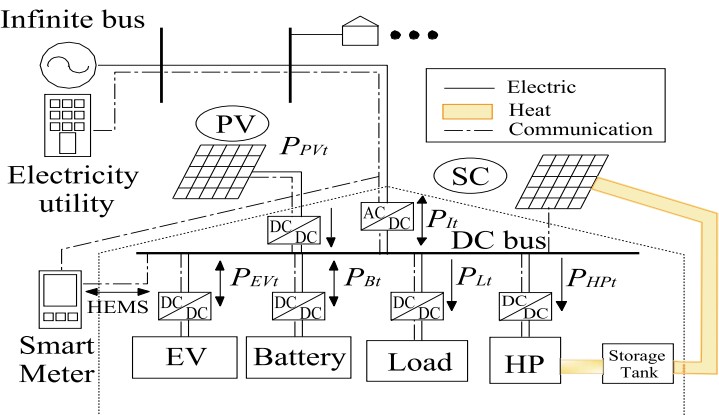

**Figure 1.** Smart home system model. EV, electric vehicle; HP, heat pump; PV, photovoltaic generation; SC, solar collector.

Moreover, the capacity of the fixed battery is 1 kW/3 kWh and the capacity of EV is 3 kW/16 kWh. The HP can supply the storage tank of the SC system with heat. Specifications of the HP is the standard type which is a 370 L type with 1.0 kW/4.0 kW heater and a coefficient of performance (COP) of 4.0 [16].

### 2.2. Photovoltaic System

The power output of PV generation $P_{PV}$ can be generally calculated by following equation [1,3].

$$P_{PV} = \eta_{PV} n_{PV} S_{PV} I_{PVI} (1 - 0.005(t_{CR} - 25)) \text{ [kW]} \tag{2}$$

where, $\eta_{PV}$ is the conversion efficiency of the solar cell array (14.4%), $n_{PV}$ is the number of panels (18 panels), $S_{PV}$ is the array area (1.3m$^2$), $I_{PVI}$ is the solar irradiation on an inclined surface (kW/m$^2$) and it is estimated using a method proposed by Erbs [17], $t_{CR}$ is the outside air temperature (°C). The rated output is 3.5 kW and the surface slope from the horizontal of the panel is 30 degrees. It is assumed that the weather condition of simulation is performed in summer season in Okinawa of Japan where the temperature is relatively warm and the solar intensity is vigorous. The weather data of Japan Meteorological Agency is utilized in the simulation.

*2.3. Solar Collector System*

The SC system is equipped with the HP which has the function of heating source. The water temperature in the storage tank is adjusted by diluting it with cooling water, which is then provided to the shower users. If the water temperature in the storage tank is lower than 40 °C in the utilization time, the water in the storage tank is preliminarily heated by the HP so that it does not fall below 40 °C. In order to calculate the operation time of the HP, thermal model which dynamic characteristic of the water temperature is considered, must be needed in the SC system. In this paper, we utilize a thermal model which the thermal dynamic characteristic including the temperature change is expressed by mathematical model, which have been consolidated by the authors in our previous work [1,3,18].

The operation time of the HP is decided as follow. At first, the amount of heat collected by the SC is calculated by the thermal model based on insolation forecast data. If the thermal energy supplied from solar radiation is not enough and the water temperature is below the goal temperature, then the operation time which HP should do, is calculated by the thermal model [1,3,18] where the thermal dynamic characteristic such as the heat loss caused by the temperature deference between inside of the tank and outside of it, is considered in detail. The thermal energy supplied by the HP can be obtained by the following equation.

$$Q_e = c\rho v_w \beta (T_e - T_h) \qquad \text{[W]} \qquad (3)$$

where, $c$ (J/(kg·K)) is specific heat of water ($c = 4.19 \times 10^3$ J/(kg·K)), $\rho$ (kg/m$^3$) is density of water ($\rho = 975$ kg/m$^3$), $v_w$ (m$^3$/h) is the quantity of hot water heated by hour, $\beta$ (Wh/J) is parameter for unit conversion ($\beta = 1/3600$ Wh/J), $T_e$ (K) is the goal temperature of the hot water in the storage tank, $T_h$ (K) is the water temperature in the storage tank.

Regarding specification of the HP, we assumed that the rated power consumption of the HP is 1 kW with on-off type in reference to the literature [19] and the COP is constant value as can be seen in the literature [20] for the simplification of the model. Generally, it is known that the COP of HP changes according to outside temperature and decreases at low temperature in winter season. In this paper, the simulation is conducted in summer season with using data of Okinawa in Japan which the climate is relatively warm throughout one years. Specification of the SC system including the HP, is shown in Table 1 [16].

**Table 1.** Specification of solar collector (SC) system.

| **Solar Heater Collector** | | |
|---|---|---|
| Heat collection efficiency | $\eta_h$ | 60 % |
| Heat collection area of one panel | $A_c$ | 1.655 m$^2$ |
| Number of panels | $n_{sc}$ | 3 panels |
| **Heat Pump** | | |
| Rated power consumption | $P_{RHP}$ | 1 kW |
| COP | $C_{cop}$ | 4.0 |
| Capacity of water storage tank | $V_w$ | 370 L |

## 3. Formulation of Optimization Problem

In this section, an optimal scheduling decision of controllable loads in the smart home is described. The objective function involving the constraint conditions is explained in Section 3.1, the optimization methodology is described in Section 3.2. The TS is explained in Section 3.3, the more detailed optimization procedure with the TS is described in Section 3.4 and the insolation forecasting method is explained in Section 3.5.

### 3.1. Set-Up of Objective Function

The objective of this research is to minimize the expected operational cost in a day for the uncertainties of the forecasted error. The objective function and constraints are explained as follow.

Objective function:

$$
\begin{aligned}
Min\; E_{day} &= E_{ELE} + E_{DEV} \\
&= \sum_{s=1}^{S} P^s \sum_{t=1}^{T} \left\{ C_t(P_{It} + \Delta P_{It}^s) + D_t|B_{It} - (P_{It} + \Delta P_{It}^s)| \right\}
\end{aligned}
\tag{4}
$$

The objective function (4) consists of two indexes. The first term $E_{ELE}$ represents conventional electricity payment which increases in proportion to energy usage amount, the electricity payment is obtained by multiplying the energy usage amount in a day (kWh) by the unit price $C_t$. In this paper, we set the unit price referencing one used in Tokyo Electric Power Company (TEPCO) [21]. The second term $E_{DEV}$ represents the penalty cost which increases according to quantity of deviation power, which have been proposed in the literature [1,3] by our previous work and it has been verified that suppression of power flow can be reduced by the electricity price system.

The penalty cost is decreased if the power flow $P_{It}$ in the smart home lies within a bandwidth of a commanded power flow which power company sets. In this paper, it is assumed that the commanded power flow from the power company to residential side has been determined throughout electricity market between power generation side and retail one such as aggregator and the power company gives the determined power flow to the residential side. The penalty cost is obtained by multiplying the deviation power (kW) by the unit price $D_t$, where the $D_t$ is varying according to the difference between commanded power flow and actual power flow in the smart home as depicted in Figure 2. For example, the penalty cost $E_{DEV}$ is calculated as follows. If the power flow exists within "region A", the penalty cost $E_{DEV}$ is calculated by multiplying the deviation power flow (kW) by the unit price $D_t = 10$ (Yen/kW). Besides, we set the constant unit price of sold power to 20 (Yen/kWh). The proposed electricity price is shown in Table 2.

The forecast error of PV output is considered in the optimization process in this paper. We consider a number of possible scenarios for the uncertainties of PV output by applying the scenario-based approach where the scenario is produced by adding a forecast error based on normal distribution. Thus, the actual power flow changes in scenario $s$ because of the forecast error and the actual power flow becomes $P_{It} + \Delta P_{It}^s$. Hence, the smallest expected cost obtained as a result of the optimization process indicates that the solution of scheduling of the controllable loads such as batteries is effectual for all scenarios.

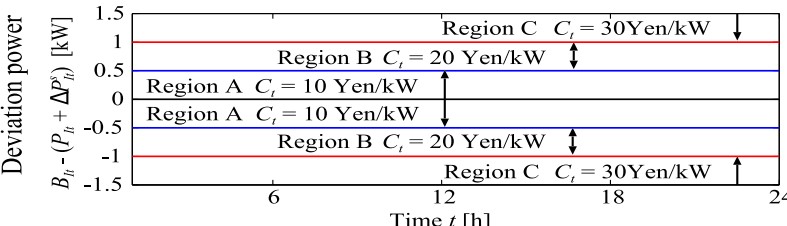

**Figure 2.** Proposed electricity price system.

Constraints conditions in solving the optimization problem to reveal the optimal scheduling of the controllable loads, are as following. Equations (5) and (6) mean inverter capacity constraints for the fixed battery and EV, respectively, these are $P_{Bmax} = 1$ kW and $P_{EVmax} = 3$ kW. Equations (7) and (8) indicate state of charge constraints for the battery and EV respectively where $C_{Bmin} = 20\%$, $C_{EVmin} = 20\%$, $C_{Bmax} = 100\%$ and $C_{EVmax} = 100\%$, respectively. Equation (9) indicates constraint for energy of battery in order to use at next day. Equation (10) indicates constraint for remaining energy of EV in preparation to be used outside and it is set to be higher than 90% of capacity of EV.

**Table 2.** Proposed electricity price.

| $E_{ELE}$ : **Electricity charge on energy usage amount** | | |
|---|---|---|
| Electricity charge | $\sum_{t=1}^{T} C_t (P_{It} + \Delta P_{It}^s)$ | In proportion to energy usage amount |
| Unit price | $C_t = 9.52$ Yen/kWh | |
| $E_{DEV}$ : **Electricity charge on deviation from power command** | | |
| Electricity charge | $\sum_{t=1}^{T} D_t \lvert B_{It} - (P_{It} + \Delta P_{It}^s) \rvert$ | In proportion to deviation power |
| Unit price | $D_t = 10$ Yen/kW | Region A |
| | $D_t = 20$ Yen/kW | Region B |
| | $D_t = 30$ Yen/kW | Region C |

Constraints:

$$|P_{Bt}| < P_{Bmax} \tag{5}$$

$$|P_{EVt}| < P_{EVmax} \tag{6}$$

$$C_{Bmin} < C_{Bt} < 0.8\, C_{Bmax} \tag{7}$$

$$C_{EVmin} < C_{EVt} < 0.8\, C_{EVmax} \tag{8}$$

$$0.5\, C_{Bmax} < C_{B(t=24)} < 0.7\, C_{Bmax} \tag{9}$$

$$0.9\, C_{EVmax} < C_{EV(t=7)} < C_{EVmax} \tag{10}$$

where,

| | |
|---|---|
| $t$: | Index for time (20 min time step) |
| $T$: | Total schedule hours ($T$ = 24 in this paper) |
| $s$: | Individual scenario ($S$ = 100) |
| $P^s$: | Probability in scenario $s$ |
| $E_{day}$: | Expected operational cost in a day |
| $E_{ELE}$: | Expected electric charge amount in a day |
| $E_{DEV}$: | Expected penalty charge amount in a day |
| $C_t$: | Unit price on electric charge |
| $D_t$: | Unit price on penalty charge |
| $B_{It}$: | Command value of power flow to smart home |
| $P_{It}$: | Power flow from power system to smart home |
| $\Delta P_{It}$: | Variation of power flow caused by forecasted error |
| $P_{Bt}$: | Charge/discharge power of fixed battery |
| $P_{EVt}$: | Charge/discharge power of EV |
| $P_{Bmax}$: | Maximum of charge/discharge power in fixed battery |
| $P_{EVmax}$: | Maximum of charge/discharge power in EV |
| $C_{Bt}$: | State of charge of fixed battery in hour $t$ |
| $C_{EVt}$: | State of charge of EV in hour $t$ |
| $C_{Bmax}$: | Maximum value of fixed battery capacity |
| $C_{EVmax}$: | Maximum value of EV capacity |
| $C_{Bmin}$: | Minimum value of fixed battery capacity |
| $C_{EVmin}$: | Minimum value of EV capacity |

### 3.2. Optimization Methodology

In this paper, a meta heuristic optimization technique is employed utilizing the TS [22,23] in order to achieve the optimal solution based on objective function (4) satisfying the constraints conditions (5)∼(10). The TS is widely employed for solving the optimization problem such as scheduling problems, with relatively shorter calculation time than the genetic algorithm (GA) [24–26]. In this work, the simultaneous re-forecast for PV output and plan controllable loads are executed every 3 h; the optimization problem can be effectively solved by the TS within a short period. Also, uncertainties of PV output are considered using a scenario-based approach which is widely employed for solving the scheduling problem and expansion planning problem of transmission network including uncertainties in power system [7–9,27,28]. The PV output can be expressed in several scenarios where each scenario is derived by adding the forecast error based on normal distribution into forecasted value in the optimization process. The solutions are operated for the possible scenarios and evaluate with expected value in the iterative step of TS optimization. Furthermore, the TS algorithm and optimization procedure are described in Sections 3.3 and 3.4, respectively.

After the optimization using TS, the optimal scheduling obtained from the optimization is operated for the real PV output, including uncertainties. This obtained simulation result is described in Section 4. In order to verify the usefulness of the proposed scheme, the scheduling of controllable loads obtained by the proposed method is tested by Monte Carlo simulation where the uncertainty of PV output expressed as possible many scenarios which are derived based on normal probability distribution.

### 3.3. Tabu Search

The TS is one of a meta heuristic global optimization method and it is discovered by Glover [29]. The TS has been effectively utilized for a combined optimization problem such a scheduling one. The problem to minimize $f(x)$ can be formulated as bellow where $x$ indicates the optimal solution to minimize the function $f(x)$ [22,30].

$$Minimize \ f(x) \tag{11}$$

$$Subject \ to \ x \in X \tag{12}$$

The Equation (11) indicates that minimization of the function of $f(x)$ provided that constraint (12) is satisfied. As first step in the TS, neighborhood solutions $x_i^*$ which is slightly moved from a present solution $x_{pre}$, are produced. $x_i^*$ which is the best solution in the neighborhood solutions, is picked out and next neighborhood solution $x_{i+1}^*$ is derived from the best solution in previous time. TS can arrive at the optimal solution by executing the iteration step until a specific criterion is satisfied. Tabu list which has function of memory system is used in order to prevent same loop which is often arising from simple iteration. The use of the tabu list which the latest moves are recorded, can decrease the possibility of remaining unsolved in a local loop. The highest evaluated solution checked in the tabu list in the neighborhood ones, is chosen as the next solution $x_{nex}$.

The implementation parameters which are employed for the TS, are as follows. The maximum value of global iteration is 2000 and the number of the tabu list is 500 and the considered number of scenarios is 100. The TS algorithm is embedded into the program to solve the optimization problem. The simulation is implemented on a desktop computer with a 2.20-GHz Intel(R) Xeon(R) E5-2660 processor with 128 GB RAM using MATLAB (R2018b).

The detail procedure of the optimization for determining the optimal scheduling of batteries and HP using the TS algorithm is described in Section 3.4.

### 3.4. Optimization Procedure

In this sub-section, the procedure of the proposed optimization method to determine the optimal schedule of controllable loads, is explained. In this paper, it is assumed that power consumption except for controllable loads and heat load which the residential people use can be forecasted. Thus, the variable number to be solved, are charge/discharge power of the fixed battery, charge/discharge of EV and operation starting time of the HP. The optimization step with using TS is as follows.

Step 1   The initial values for charge/discharge power of the battery and EV are set in addition to the operation starting time of the HP.

Step 2   The neighborhood solutions which is slightly moved from initial solution or the chosen solution in a previous iteration, are produced and evaluated.

Step 3   The neighborhood solutions are evaluated in the Equation (4), the best neighborhood solution which is not recorded is chosen and recorded in the tabu list. If the best solution already lies in the tabu list, the next best solution in the neighborhood ones is selected. Even if the selected solution is worse than a solution selected in the previous iteration, its registration to the tabu list is executed. Old recorded solutions are overwritten with the new chosen one in turns and new solutions are recorded.

Step 4   If the chosen solution is better than the solution which is previously reserved as the optimal solution, the chosen solution is reserved as the optimal one.

Step 5   If number of the global iteration achieves criteria one which was set in advance, the search process is finished. Otherwise, the algorithm proceeds to Step 6.

Step 6   For the best solution obtained in Step 3, the process goes to Step 2 where the neighborhood solutions are derived from the best one again.

where, the pseudo-code of the optimization algorithm using TS is as follows Algorithm 1 [22].

---

**Algorithm 1** Optimization algorithm using TS

---

$x = x_0$; % Set of initial solution
**for** $j = 1$ to $N_S$ **do**

   Generate a set $A$ which is a neighborhood solutions from $x$ or previous one $x_{nei_N}$;
   Find best neighborhood solution $x'$ of $A$; % check of constraints conditions, the solutions is not recorded in the tabu list;
   $x = x'$;
   Update tabu list;
   **if** $f(x) < f(x_{BEST})$ **then**

      $x_{BEST} = x$
      $x_{nei_{N+1}} = x$
   **end if**
**end for**
$x_{BEST}$ is the best-found solution.

---

### 3.5. Insolation Forecasting Method

A NN is widely employed as one of the forecasted methods for insolation [31–33]. In this paper, an NN is used for the insolation forecasting method. Parameters entered into the NN toolbox in MATLAB® are depicted in Table 3. Input factors such as a data of atmospheric, temperature, humidity and hours of sunlight for the previous 24 h are entered into the NN. It can then forecast insolation for the proceeding 24 h. Furthermore, re-forecast and re-plan are conducted in the optimization for the scheduling of controllable loads. The data fed into the NN is updated every 3 h. The overview of re-forecast and updating data is shown in Figure 3. Learning (in which optimize synaptic weight is optimized) is carried out for 365 days. After learning of the NN, the weights are applied for another year in order to confirm the availability of the optimized synaptic weight. A probability density of the forecast error of insolation for the year is depicted in Figure 4. In addition to this, a probability density for the case of updating data is depicted in same figure. It is observed that the forecast error can be reduced by updating the data such as atmospheric and temperature which are used to forecast.

**Table 3.** Parameter entered into neural network toolbox in MATLAB$^{\circledR}$.

| Number of learning | 30,000 |
| --- | --- |
| Termination condition of error | $1.0 \times 10^{-4}$ |
| Learning rate | 0.1 |
| Algorithm of updating weight | Steepest descent method |
| Input/output function | Sigmoid function |
| Input factor | Atmosphere, Temperature, Humidity, Hours of sunlight |

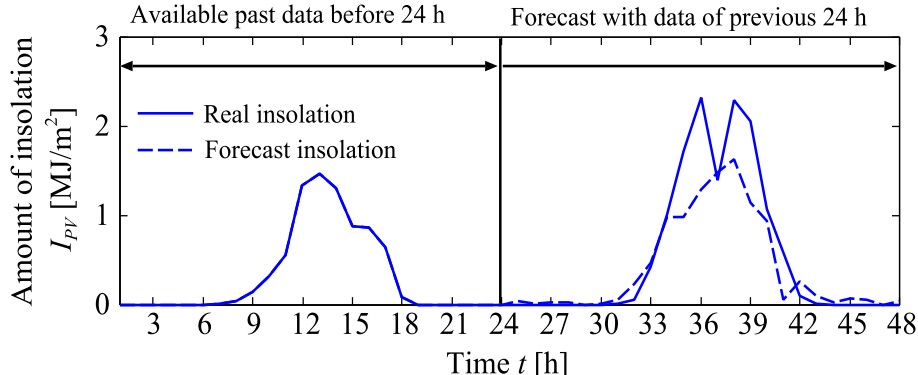

(**a**) Amount of insolation without updating of data.

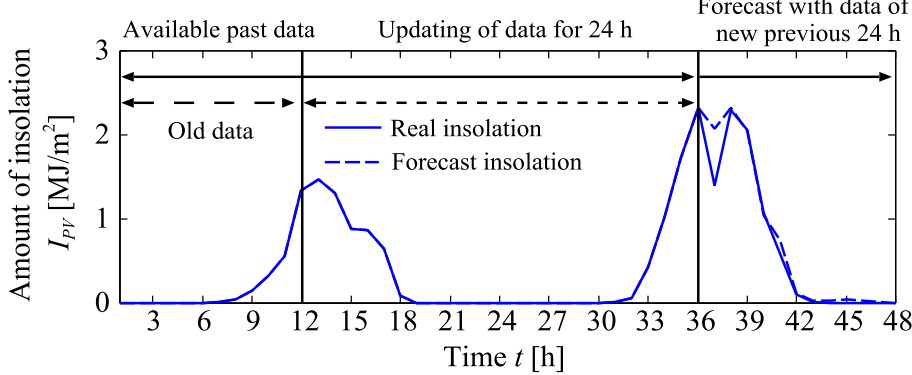

(**b**) Amount of insolation with updating of data.

**Figure 3.** Overview and illustration of re-plan and re-forecast.

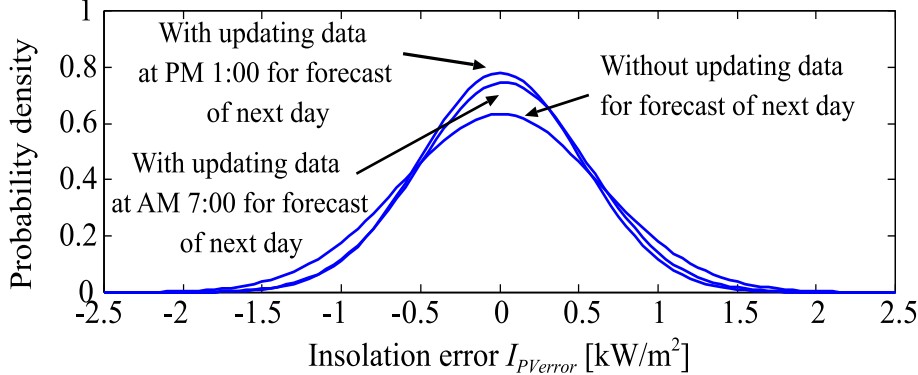

**Figure 4.** Forecast error for insolation with neural network (NN).

The whole flow chart of optimization including the forecasting with the NN and consideration of the uncertainties is depicted in Figure 5. At first, PV output is obtained in forecasting the insolation with NN. Next, the forecasted error based on Figure 6 is added into the forecasted PV output. In this paper, scenario-based approach is applied for the uncertainties. A number of scenarios are derived by adding forecast error which is expressed as shown in Figure 6. In process step 3~step 6, the neighborhood solutions are tried to operate for all scenarios and evaluated by the objective function Equation (4) until the global iterations are satisfied. In this paper, re-forecast and re-plan are considered. Thus, the schedule once planned is renewed every 3 h by using the latest data in the optimization process with the TS algorithm as shown in step 7. Hence, the re-forecast and re-plan are executed for 8 times in a day.

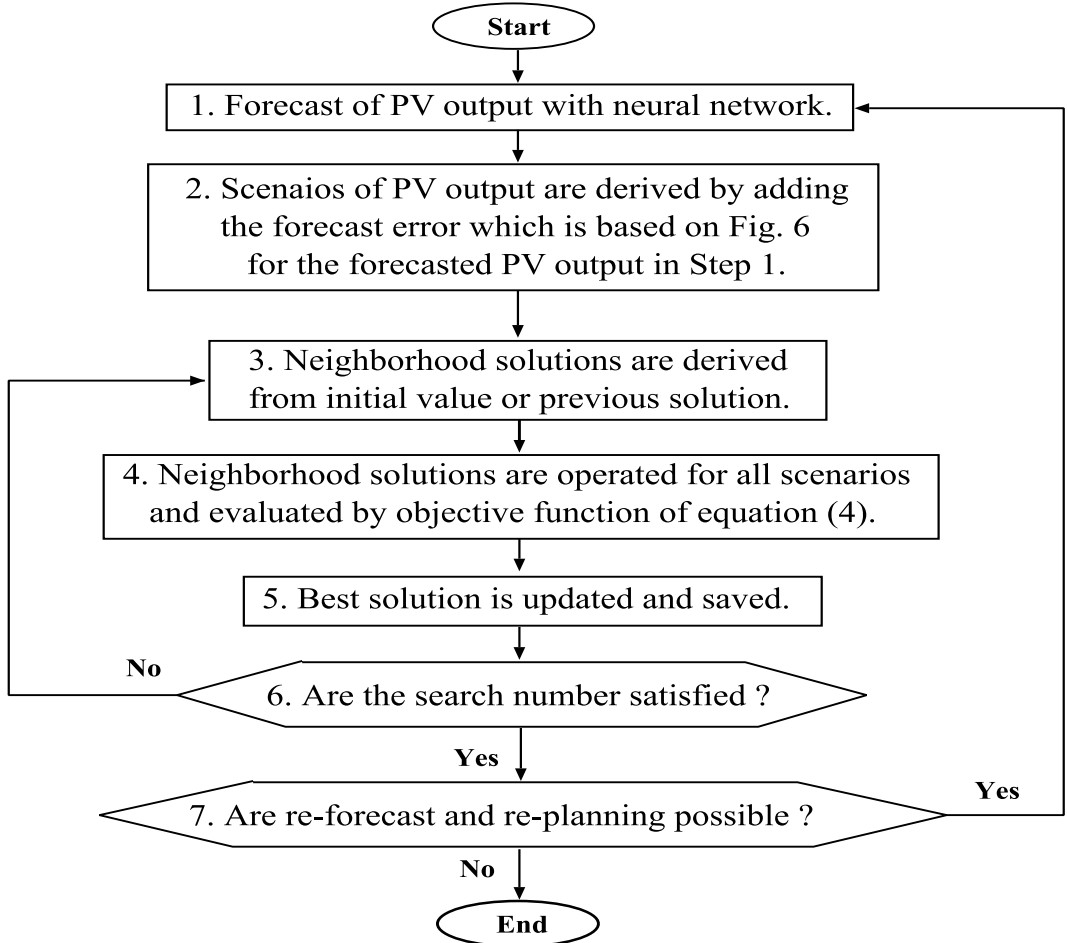

**Figure 5.** Flow chart in optimization.

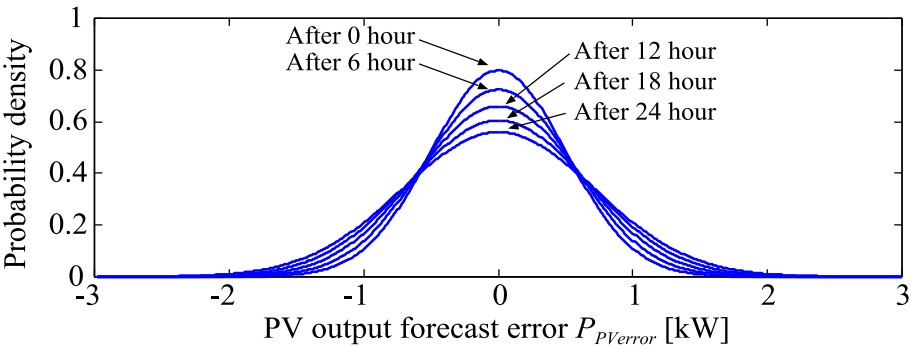

**Figure 6.** Forecast error probability distribution for photovoltaic (PV) output power.

## 4. Simulation Results and Discussion

In this section, simulation results are described in order to validate the effectiveness of the proposed method. In Section 4.1, simulation conditions are presented. The simulation results and the statistical analysis are presented in Sections 4.2 and 4.3, respectively.

### 4.1. Simulation Conditions

Preconditions in the simulation are as follows.

1.  It is supposed that power consumption and heat load in the smart home can be forecasted. The assumed power consumption of uncontrollable loads is shown in Figure 7a. NN is used for forecasting the insolation and the PV output which is calculated by Equation (2) from forecasted insolation, is depicted in Figure 7b where the forecasted error based on Figure 6 is added into the forecasted PV output.
2.  In regard to the heat load, which is as follows. Three people use 30 L and 150 L hot water as shower from 7:00 to 8:00 and 19:00 to 22:00. When the water temperature in the storage tank is lower than 40 °C at 8:00 and 22:00, the water is heated by the HP.
3.  Normal people often use EV as drive mode for a few hours of the day. Thus, we assume that the EV is used in drive mode from 8:00 to 18:00 and discharges 700 W/h outside. Except, for the drive time the EV connected to the smart home and has the function of both vehicle-to-grid (V2G) and grid-to-vehicle (G2V).

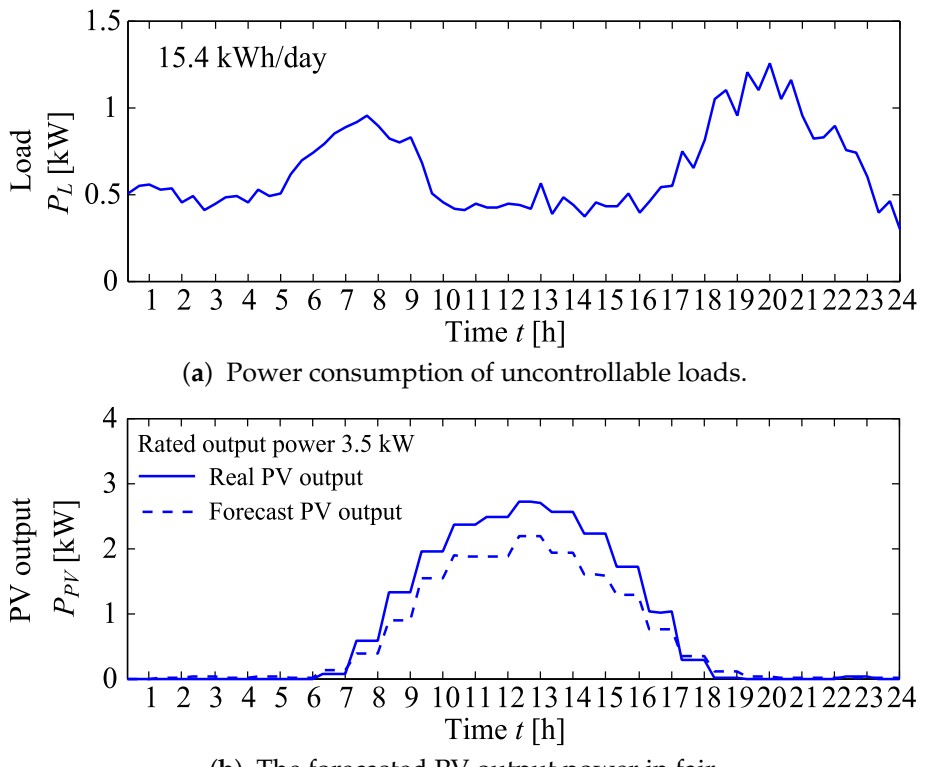

(**a**) Power consumption of uncontrollable loads.

(**b**) The forecasted PV output power in fair.

**Figure 7.** Power consumption and PV output power.

### 4.2. Simulation Results

In order to verify the effectiveness of the proposed method, the simulation results obtained in four different cases are compared and discussed in this sub-section. Case 1 is that the optimization is executed using the data only in the previous day for forecasting PV output in next day in which re-forecast, re-plan and uncertainties are not considered. Case 2 is that the optimization considering

uncertainties of PV output is executed but re-forecast and re-plan are not considered in the case. Case 3 is that the optimization considering re-forecast and re-plan is executed but uncertainties for PV output are not considered in the case. Case 4 is that all (uncertainties, re-forecast and re-plan) in former ones are considered in the optimization.

The simulation result in the case 1 is depicted in Figure 8. This figure shows that result which the scheduling obtained by the optimization without considering re-forecast and re-plan, was operated for the real PV output as shown in Figure 7b. The temperature in a storage tank in case is depicted in Figure 8a. It can be seen from these figures, the temperature in the storage tank does not fall 40 °C at 8:00 and 22:00 by heat supply from SC and the HP. The power consumption of the HP is depicted in Figure 8b. It is confirmed that the HP operates before at 6:00, since the temperature in the storage tank falls to 40 °C at 7:00 without heating from the HP. As respects the use of heat water at night time, the temperature in the storage tank does not fall 40 °C without heating from the HP after at 22:00 because much insolation in the daytime in fair day was obtained and SC could supply the storage tank with collected heat. It can be seen that the operation time of the HP is minimized with the maximized utilization of solar energy.

Next, the power flow of interconnected point in the smart home in case 1 is indicated in Figure 8c. It can be observed that the power flow deviates from bandwidth given by a power company due to the forecasted error in which re-forecast, re-plan and uncertainties are not considered. Furthermore, the state of charge for the battery and EV in case 1 is shown in Figure 8d, the charge/discharge power of the batteries is performed within the allowed constraints conditions.

The simulation result in the case 2 is depicted in Figure 9. This figure shows that result which the scheduling obtained by the optimization with considering uncertainties except for consideration of re-forecast and re-plan, was operated for the real PV output. The temperature in the storage tank does not fall 40 °C at 8:00 and 22:00 by heat supply from SC and the HP as can be seen from Figure 9a,b. It can be seen from Figure 9c, the deviation from bandwidth of power flow is improved in comparison with case 1 however, it can be observed that the deviation from bandwidth of power flow exist at around 17:00 a little. Additionally, the power flow exist in the vicinity of the bandwidth limit in the daytime.

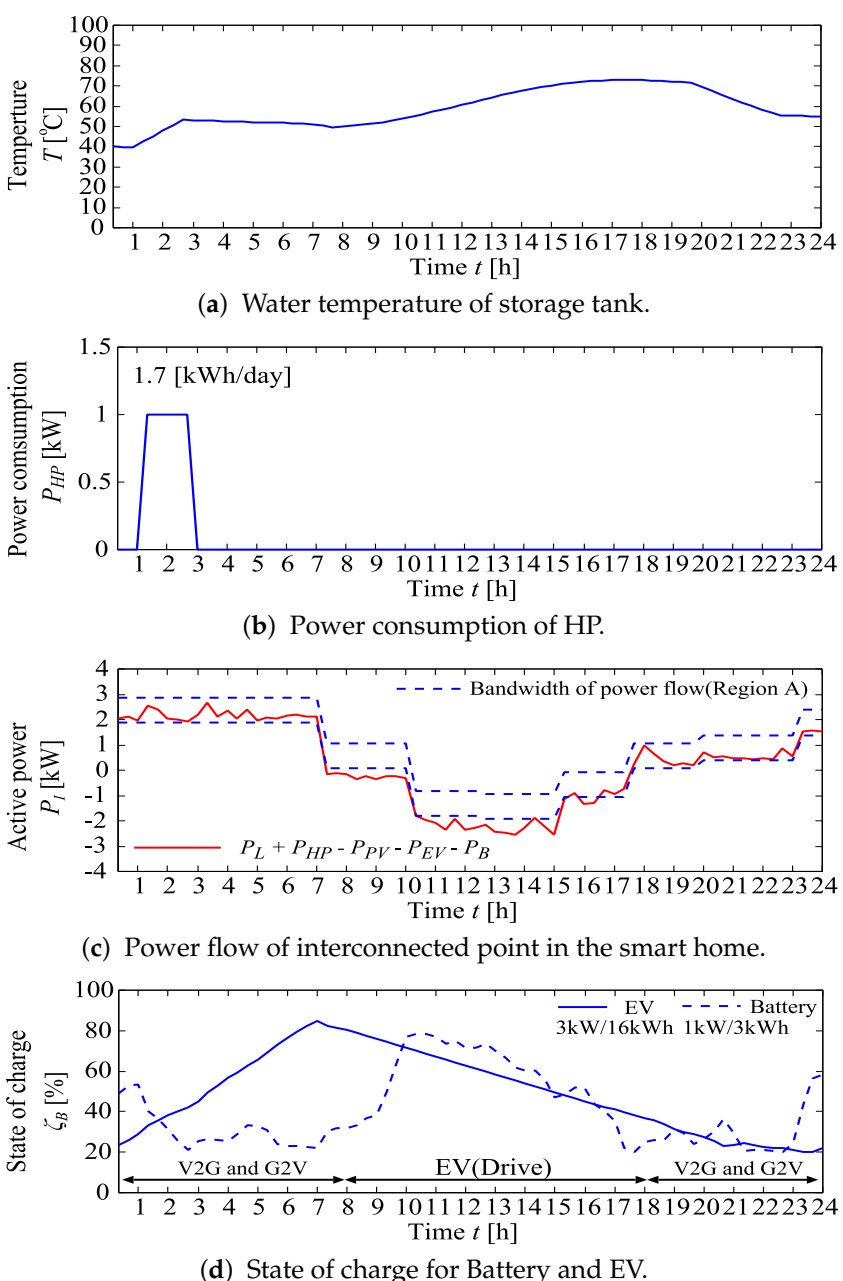

(**a**) Water temperature of storage tank.

(**b**) Power consumption of HP.

(**c**) Power flow of interconnected point in the smart home.

(**d**) State of charge for Battery and EV.

**Figure 8.** Simulation results without considering re-forecast, re-plan and uncertainties on a fair day (Case 1).

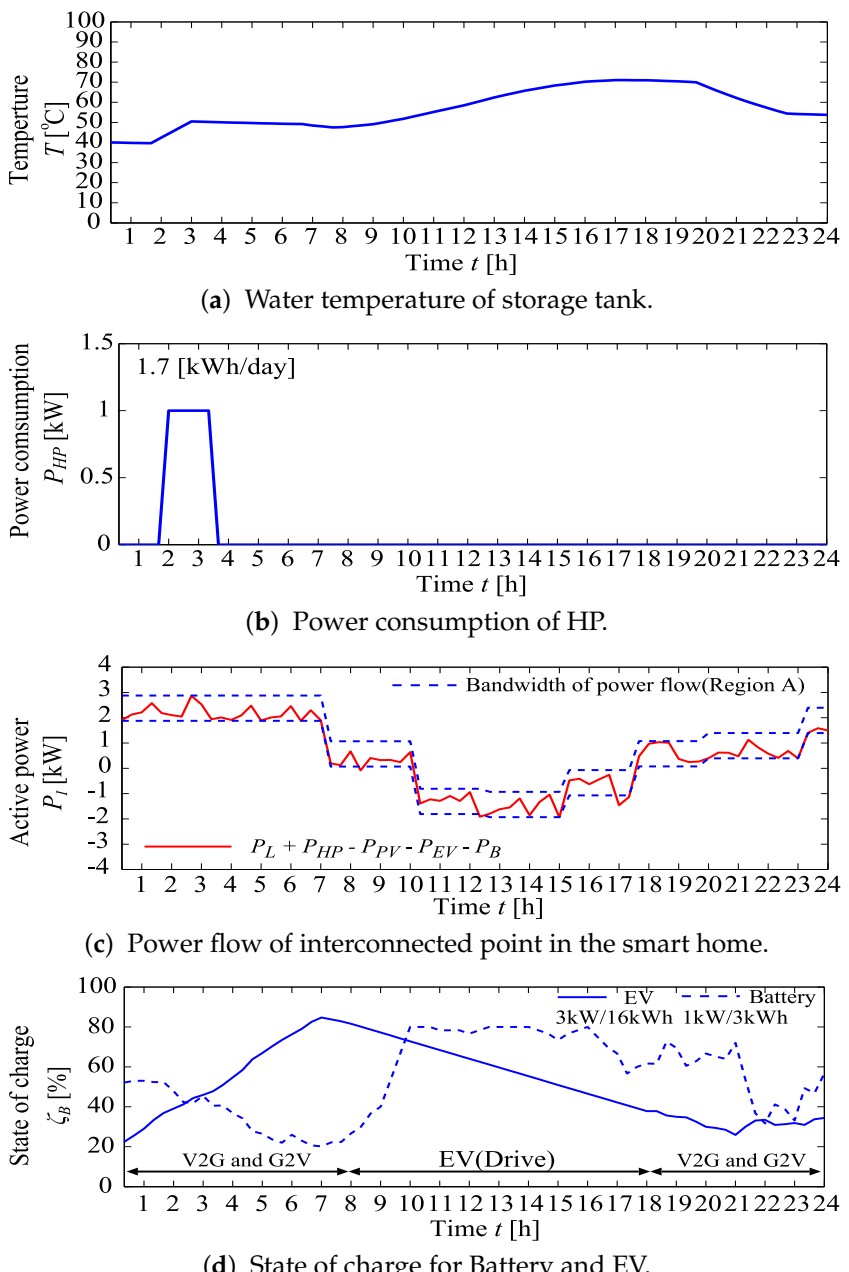

(**a**) Water temperature of storage tank.

(**b**) Power consumption of HP.

(**c**) Power flow of interconnected point in the smart home.

(**d**) State of charge for Battery and EV.

**Figure 9.** Simulation results with considering uncertainties, without considering re-forecast and re-plan on a fair day (Case 2).

The simulation result in the case 3 is depicted in Figure 10. This figure shows that result which the scheduling obtained by the optimization with considering re-forecast and re-plan except for uncertainties, was operated for the real PV output. It can be seen that the heat demand is satisfied as shown in Figure 10a,b as well as case 1 and case 2. It can be seen from Figure 10c, the deviation from bandwidth of power flow is improved in comparison with case 1 however, in this case, it can be observed that the deviation from bandwidth of power flow partially still exist at around 8:00∼10:00 a little.

The simulation result in the case 4 is shown in Figure 11. This figure shows that result which the scheduling obtained by the optimization with considering uncertainties, re-forecast and re-plan, was operated for the real PV output. It can be seen that the heat demand is satisfied as shown in Figure 11a,b as well as case 1∼case 3. It can be seen from Figure 11c, the deviation from bandwidth of power flow exists for a whole day although power flow in case 2 and case 3 partially deviates

from the bandwidths for a day. In case 4, the scheduling of controllable loads is determined by the optimization so that the operational cost is minimized for many scenarios which the forecast error derived from the probability density distribution shown in Figure 6, is added for the forecasted PV output. Consequently, this application of the optimization method contributes to establishment of robust scheduling of controllable loads for uncertainties.

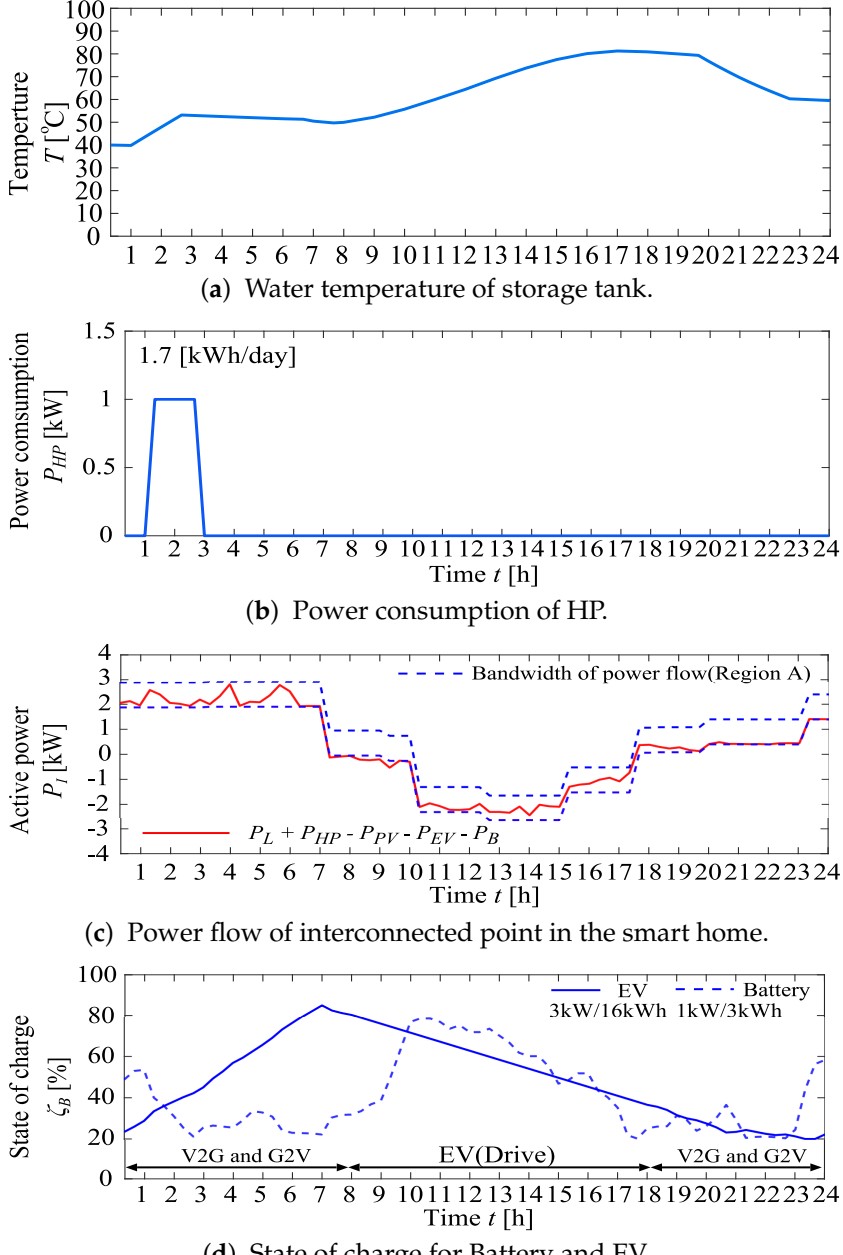

(**a**) Water temperature of storage tank.

(**b**) Power consumption of HP.

(**c**) Power flow of interconnected point in the smart home.

(**d**) State of charge for Battery and EV.

**Figure 10.** Simulation results with considering re-forecast and re-plan, without considering uncertainties on a fair day (Case 3).

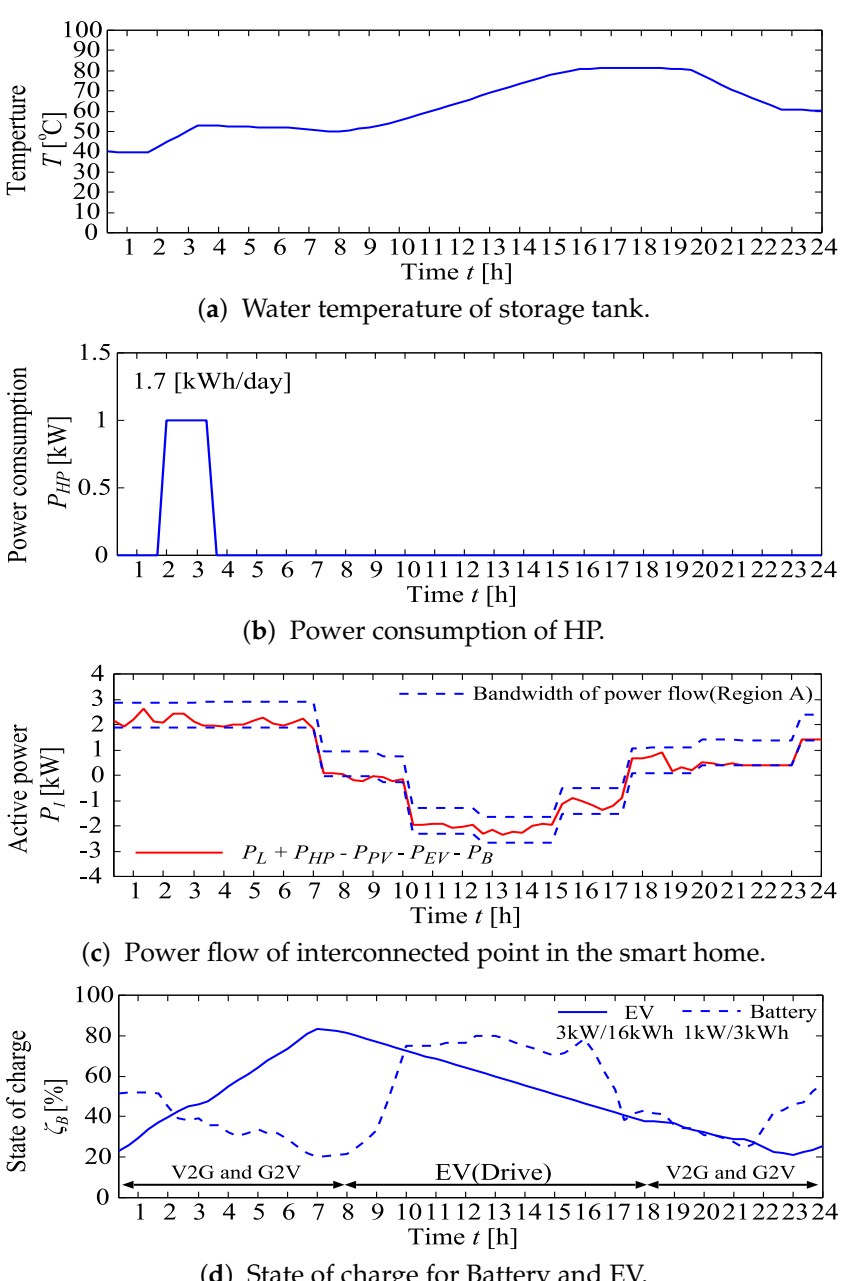

(**a**) Water temperature of storage tank.

(**b**) Power consumption of HP.

(**c**) Power flow of interconnected point in the smart home.

(**d**) State of charge for Battery and EV.

**Figure 11.** Simulation results with considering re-forecast, re-plan and uncertainties on a fair day (Case 4).

### 4.3. Statistical Analysis

In order to verify the effectiveness of the proposed method, we examined the optimal scheduling of the controllable loads obtained by the optimization in case 1~case 4 how robust it is for uncertainties. Monte Carlo simulation is conducted for 1000 scenarios in order to obtain a statistically certain cost. Figure 12 indicate that the power flow when the scheduling of controllable loads obtained by the optimization in case 1~case 4 was operated for uncertain PV output. In case 1, it is obviously observed that power flow deviates from the bandwidth in the daytime affected by the PV output as shown in Figure 12a. The deviation of power flow from the bandwidth in case 2 and case 3 is reduced in comparison with case 1 by considering uncertainties or re-forecast and re-plan as can be seen from Figure 12b,c. The deviation of power flow in case 4 is further improved in comparison with case 2 and case 3 by incorporating the scenario-based approach, re-forecast and re-plan into the optimization process.

Figure 13 and Table 4 show distribution and histogram of the operational cost and the expected cost obtained by Monte Carlo simulation for 1000 scenarios which are derived by adding possible forecast error based on the normal distribution to the forecasted PV output. The verification results indicate that the expected value of operational cost for case 1 in which both re-forecast, re-plan and uncertainties are not considered, is the most expensive as depicted in Figure 13a (482 Yen/day). Furthermore, in case 2 which uncertainties are only considered, the cost is 456 [Yen/day] as depicted in Figure 13b. In case 3 which re-forecast and re-plan are only considered, the cost is 426 [Yen/day] as depicted in Figure 13c. The cheapest cost is depicted in case 4 which re-forecast, re-plan and uncertainties are considered as shown in Figure 13d (412 Yen/day).

More discussion for the simulation results between case 1 and case 4, is as follows. if the real PV output would come out exactly as the forecasted one, the operational cost is minimized by operating the optimal scheduling of controllable load obtained by the optimization in even case 1. It means that the power flow as shown in Figure 8c exists in the low position of the bandwidth in Region A so that power consumption is minimized based on set the objective function. However, real PV output often deviates from the forecasted one. Thus, the controllable loads should be scheduled while considering the uncertainties in advance. That is the power flow shown in Figure 11c exists in the center within the bandwidth in the day time in preparation for the capability in which the forecast error would be relatively higher throughout the day.

The frequency of the state of charge on the battery for 1000 scenarios in case 1 and case 4 are depicted in Figure 14a,b, respectively. The state of charge on the battery exists at almost 80% from 10:00 to 24:00 in case 1 which re-forecast, re-plan and uncertainties are not considered since the battery acts at the maximum of its capacity without consideration of the forecast error of PV output as shown in Figure 14a. On the other hand, in case 4 which re-forecast, re-plan and uncertainties are considered, the state of charge on that battery exists at almost 40~60% from 10:00 to 24:00 in preparation for the forecast error of PV output as shown in Figure 14b. These results imply that the capacity of the battery may be even cut down by introducing the proposed method.

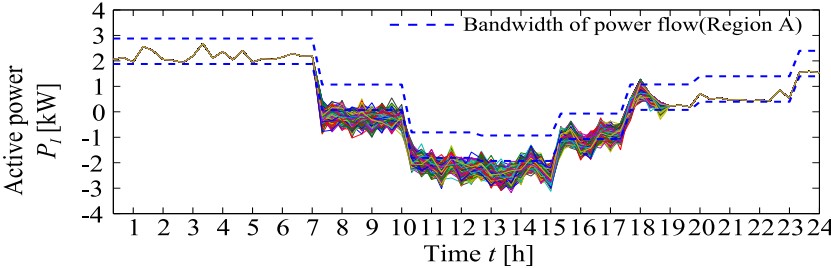

(**a**) Power flow of interconnected point in the smart home
for uncertain PV output (in the possible 1000 scenarios) in case 1.

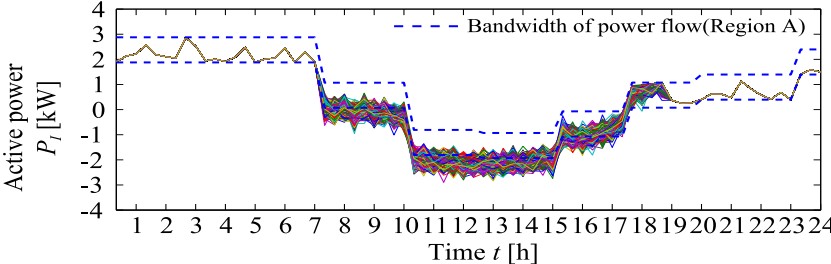

(**b**) Power flow of interconnected point in the smart home
for uncertain PV output (in the possible 1000 scenarios) in case 2.

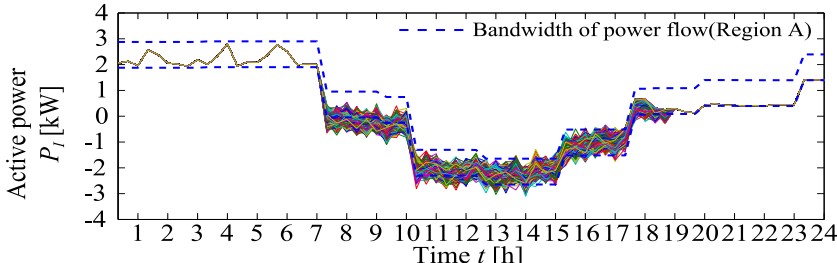

(**c**) Power flow of interconnected point in the smart home
for uncertain PV output (in the possible 1000 scenarios) in case 3.

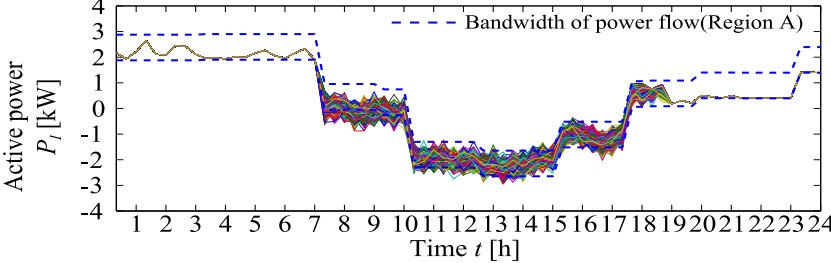

(**d**) Power flow of interconnected point in the smart home
for uncertain PV output (in the possible 1000 scenarios) in case 4.

**Figure 12.** Statistical analysis result in regards to power flow in the smart home in each case.

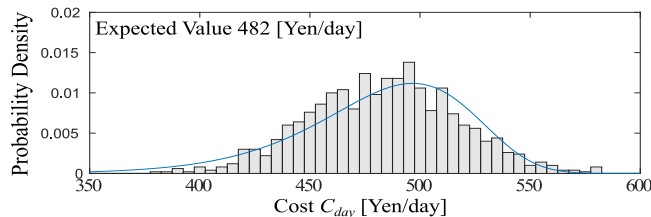

(**a**) Distribution of operational cost in case 1
(without considering re-forecast, re-plan and uncertainties).

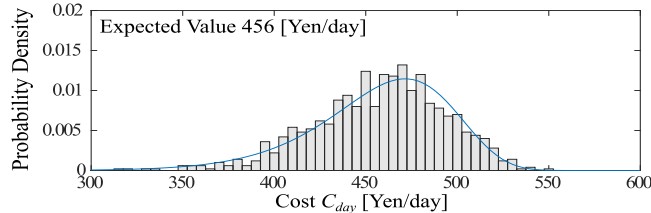

(**b**) Distribution of operational cost in case 2
(with considering uncertainties, without considering re-forecast and re-plan).

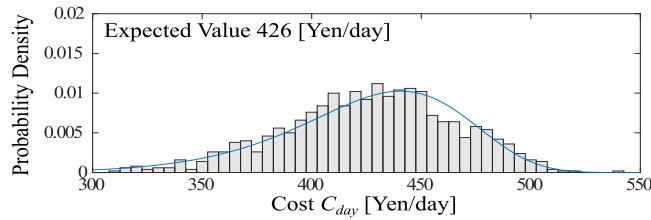

(**c**) Distribution of operational cost in case 3
(without considering uncertainties, with considering re-forecast and re-plan).

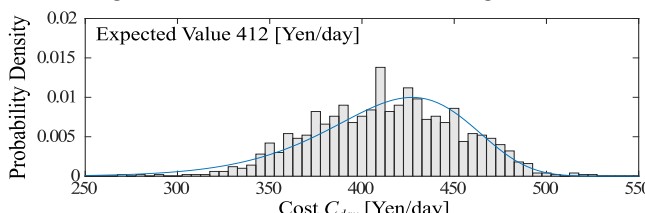

(**d**) Distribution of operational cost in case 4
(with considering re-forecast and re-plan, with considering uncertainties).

**Figure 13.** Histogram of operational cost in each case.

**Table 4.** Expected operational cost for 1000 scenarios in each case.

|  | Case 1 | Case 2 | Case 3 | Case 4 |
|---|---|---|---|---|
| Consideration of uncertainties | × | ○ | × | ○ |
| Consideration of re-forecast and re-plan | × | × | ○ | ○ |
| **Expected operational cost** | 482 Yen/day | 456 Yen/day | 426 Yen/day | 412 Yen/day |

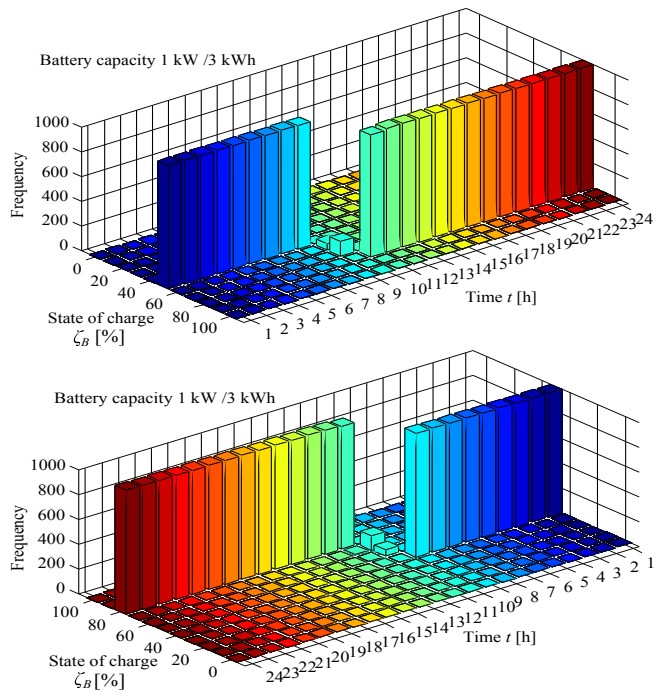

(**a**) Case 1 (without considering re-forecast, re-plan and uncertainties).

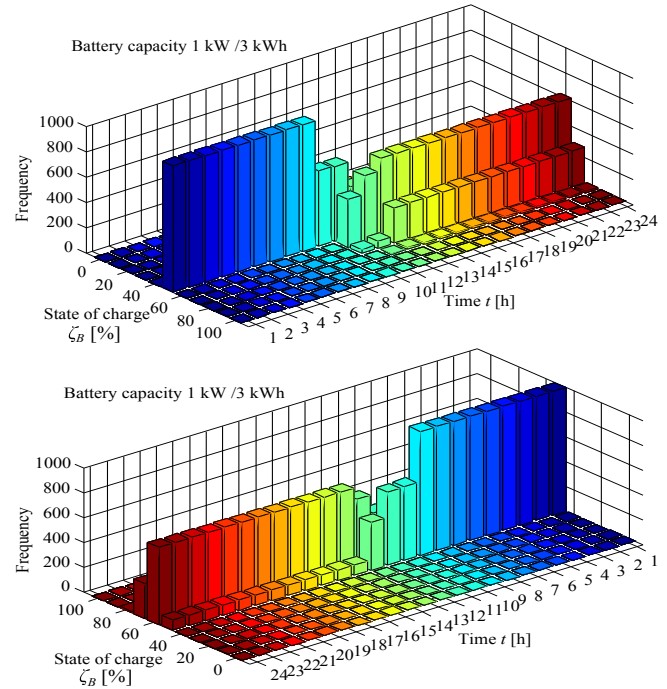

(**b**) Case 4 (with considering re-forecast, re-plan and uncertainties).

**Figure 14.** Histogram of state of charge on battery for 1000 scenarios in each case.

## 5. Conclusions

This paper proposes an optimal scheduling method of controllable loads considering uncertainties, re-forecast and re-plan. A fixed battery, EV and HP are utilized as controllable loads in order to control the power flow of interconnected point in the smart home. The optimization of determining the optimal scheduling of controllable loads is conducted by using TS. Moreover, to solve the optimization problem involving uncertainties caused by the forecasted error, a scenario-based approach is applied to the proposed scheme. The optimal scheduling is obtained from the optimization and it is observed

that the power flow in the smart house can be controlled within the given bandwidth from the power company. Furthermore, to confirm the robustness of the proposed scheme, 1000 scenarios are tested, including uncertainties of PV output using the Monte Carlo simulation. The statistical analysis shows that the expected operational cost can be reduced by applying the proposed optimization method considering the uncertainties, re-forecast and re-plan. Furthermore, the statistical analysis shows that the frequency of the state of charge on that battery considering the uncertainties, re-forecast and re-plan exists at almost 40~60% from 10:00 to 24:00 although the frequency of that without considering uncertainties, re-forecast and re-plan exists at almost 80% in the same period. The statistical analysis indicates that there is the possibility of battery capacity reduction by introducing the proposed method.

**Author Contributions:** Conceptualization, A.Y.; methodology, A.Y.; resources, K.U.; data curation, M.K.; writing—original draft preparation, A.Y.; writing—review and editing, A.Y., S.C., N.K. and Z.Y.; supervision, T.S.

**Funding:** This research received no external funding.

**Conflicts of Interest:** The authors declare no conflict of interest.

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
