# Peer review of "Optimal Scheduling Method of Controllable Loads in Smart Home Considering Re-Forecast and Re-Plan for Uncertainties"

_applsci, doi:10.3390/app9194064_

Round 1

Reviewer 1 Report

It is substantially well written and presented research paper in scope of the journal. The scope of the study is clear and undertaken methodology is also novel in building simulation analysis. Although, the paper is required minor changes as follows;

-The abstract should be revised. The reason is that the explanation of the research problem is too lengthy. I recommend to the authors that highlighting only the main issue which is relevant to the scope of this paper. The aim of the research should be stated explicitly in the abstract as well. One another note that the methodology should be mentioned. In this research paper, the quantitative research method, predominantly statistical analysis and simulations are undertaken concurrently. 

-In section 5 (Conclusion) should be written substantially and supported with specific findings from the statistical graphs presented in section 4. 

Author Response

Dear Reviewer and editor

The authors are grateful to the reviewers and editor for the valuable comments that helped us significantly to improve the quality of our manuscript and clarify its superiority point.

Sincerely

Akihiro Yoza

Reviewer 2 Report

The motivation of this study in the light of considered problem statement is unclear. Kindly clarify these aspects in the introduction. The literature review is very brief and general. An in-depth review of related research is required to establish the importance of the studied objectives. In other words, the number of relevant references has to be increased. The algorithm in section 3.3 should be shown in a pseudo code format. What is the computational cost of the TS and how does it fare compared to other applicable optimization techniques. What is the duration of time step in this study? Figure 8/9 (d): EV operates in only 2 modes; V2G and Driving? In the literature, the term V2G corresponds to the discharging of EV battery energy to the grid and G2V refers to the charging. According to this figure, the charging of EV is also termed as V2G. Please chose proper naming of the EV operation principles according to the widespread literature.  The significance of the proposed method should be established by comparison with other method(s)/research paper(s). The paper needs to be thoroughly revised from English language perspective.

Author Response

(The authors gave the same response as above.)

Round 2

Reviewer 2 Report

1. The authors mention the lack of consideration of HVAC, CHP etc as controllable loads in previous research. Various studies have considered this aspect:

[A] Rafique, M.K.; Haider, Z.M.; Mehmood, K.K.; Saeed Uz Zaman, M.; Irfan, M.; Khan, S.U.; Kim, C.-H. Optimal Scheduling of Hybrid Energy Resources for a Smart Home. Energies 2018, 11, 3201.

[B] Rafique, M.K.; Khan, S.U.; Saeed Uz Zaman, M.; Mehmood, K.K.; Haider, Z.M.; Bukhari, S.B.A.; Kim, C.-H. An Intelligent Hybrid Energy Management System for a Smart House Considering Bidirectional Power Flow and Various EV Charging Techniques. Appl. Sci. 2019, 9, 1658.

2. Previous research has established optimal coordination of various energy resources of a smart home. Do the authors wish to portray that their significant contribution is to consider HP among other resources? (lines 35 to 44). Can previously proposed methodologies be applied to incorporate HP? 

3. The introduction section needs to be completely rewritten as it is very loosely organized and confuses the reader. Please format the introduction as:
- Background and problem statement
- literature review
- motivation/necessity of proposed research in light of previous research
- main objectives and contributions of this study
- organization of the manuscript

The objectives and contributions should be mentioned in the form of points for clarity.

4. The definitions of various acronyms have been randomly defined after their first time usage. Please fix this.

5. The authors should implement this study with GA (as mentioned in section 3.2) and present a comparison in the results with TS to establish the significance of their proposed method.

In the previous round, the authors were requested to establish such comparison however, it was not included in the revised version. Such comparison is vital for the publication of this study.

Author Response

I am here submitting the revised manuscript entitled

Optimal Scheduling Method of Controllable Loads in Smart Home Considering Re-forecast and Re-plan for Uncertainties,

to the MDPI Applied Sciences.

The authors are grateful to the reviewers and editor for the valuable comments that helped us significantly to improve the quality of our manuscript and clarify its superiority point.

Thanking you.

Sincerely,

Dr. Akihiro Yoza

Round 3

Reviewer 2 Report

I appreciate the revisions incorporated by the authors. The manuscript is in good organization now and suitable for publication.